# An Optimized HPLC-DAD Methodology for the Determination of Anthocyanins in Grape Skins of Red Greek Winegrape Cultivars (*Vitis vinifera* L.)

**DOI:** 10.3390/molecules27207107

**Published:** 2022-10-21

**Authors:** Natasa P. Kalogiouri, Christina Karadimou, Mary S. Avgidou, Elissavet Petsa, Emmanouil-Nikolaos Papadakis, Serafeim Theocharis, Ioannis Mourtzinos, Urania Menkissoglu-Spiroudi, Stefanos Koundouras

**Affiliations:** 1Laboratory of Analytical Chemistry, Department of Chemistry, Aristotle University of Thessaloniki, 54124 Thessaloniki, Greece; 2Laboratory of Viticulture, School of Agriculture, Faculty of Agriculture Forestry and Natural Environment, Aristotle University of Thessaloniki, 54124 Thessaloniki, Greece; 3Pesticide Science Laboratory, School of Agriculture, Faculty of Agriculture Forestry and Natural Environment, Aristotle University of Thessaloniki, 54124 Thessaloniki, Greece; 4Laboratory of Food Chemistry and Biochemistry, Department of Food Science and Technology, Faculty of Agriculture, Aristotle University of Thessaloniki, 54124 Thessaloniki, Greece

**Keywords:** anthocyanins, HPLC, hierarchical cluster analysis, positive environmental footprint, Greek grape varieties, *Vitis vinifera*

## Abstract

A rapid and simple HPLC-DAD analytical method was developed and optimized for the determination of anthocynanins in three red Greek winegrape varieties (Kotsifali, Limnio, and Vradiano). The critical parameters, such as the acidifying solvent and the extraction temperature, which affect the extraction of anthocyanins from the grapes, were studied to find the optimum values. The developed methodology was validated in terms of selectivity, linearity, accuracy, and precision and presented satisfactory results. The limits of quantification (LOQs) ranged between 0.20 mg/kg to 0.60 mg/kg, and the limits of detection (LODs) ranged between 0.06 mg/kg and 0.12 mg/kg. The RSD% of the within-day and between-day assays were lower than 6.2% and 8.5%, respectively, showing adequate precision. The accuracy ranged between 91.6 and 119% for within-day assay and between 89.9 and 123% for between-day assay. Sixteen samples from the main regions of each variety as well as from the official ampelographic collections of Greece were collected during the 2020 growing season and were further analyzed by HPLC-DAD. Notable differences in the anthocyanin content were detected among the cultivars using hierarchical cluster analysis (HCA).

## 1. Introduction

Phenolic compounds constitute a ubiquitous group of natural pigments from the flavonoid family widely distributed in fruits. These polyphenolic compounds are glycosides of polymethoxy and polyhydroxy derivatives of the 2-phenylbenzopyrylium or the flavylium ion [1]. The aglycone forms constitute the group of anthocyanidins. The most common anthocyanidins are delphinidin, cyanidin, malvidin, petunidin, peonidin, and pelargonidin. Grapes are rich in anthocyanidins with the exception of pelargonidin. Grape anthocyanidins belong to a diverse group of compounds called “secondary metabolites”, which are synthesized principally as a plant adaptation to abiotic or biotic stresses, but which are also crucial for the quality of red wines, namely wine color intensity, hue, and stability.

Anthocyanins are located in the skins of grape berries, starting their accumulation at the veraison stage, a short lag phase separating two distinct periods of berry development: an initial phase characterized by rapid cell division and expansion in green berries and a second phase of growth by active solute accumulation, corresponding to grape ripening, at the end of which the anthocyanins reach their maximum levels. Thus, anthocyanins are important indicators in the determination of the harvest date.

Several agricultural factors exert a significant effect on the levels of grape anthocyanins. Natural factors such as the topography, soil, and climate of a vineyard location are reported to have a measurable impact on grape and wine color, mostly associated with their ability to induce different levels of vine growth and yield [2]. Seasonal operations (e.g., pruning, canopy manipulation, irrigation, fertilization, floor management) also affect the levels of anthocyanins by adapting the thermal and light conditions in the vine canopy [3]. However, genotype is the main factor differentiating the anthocyanin content of grapes and wines since the natural levels in berry skins are highly variable among *Vitis vinifera* cultivars. Furthermore, the profile of anthocyanins (relative abundance of individual anthocyanins, ratio of di-oxygenated vs. tri-oxygenated side-ring forms, ratio of acylated vs. non-acylated derivatives, etc.) is also variable among grapevine varieties [4,5]; therefore, it can be used as a chemotaxonomical criterion to establish differences between *Vitis vinifera* grape varieties [6] or other *Vitis* species [7].

The chemical characterization of grapevine cultivars is particularly important for the protection of minority varieties and the conservation of genetic diversity within the *Vitis vinifera* germplasm, which is endangered due to the generalized use of a small number of grape varieties in the international wine market [8]. Characterization and utilization of these varieties could also satisfy the increasing demand for new styles of wines by wine consumers [9]. Moreover, the profile of anthocyanins has technological and organoleptic repercussions on the winemaking process since it affects the intensity and stability of the red color in wine [10,11]. Color intensity increases with the number of substituted groups on the B-ring (di-oxygenated forms are redder while tri-oxygenated shift to blue) and with the replacement of hydroxyl by methoxyl groups (i.e., malvidin has the darkest color). Methoxylated anthocyanins (malvidin and peonidin) are also more stable than hydroxylated ones to environmental and viticultural factors [12]. Thereby, both the levels and the relative proportion of different anthocyanins in grape skins can confer distinctive characteristics to the produced wines depending on the cultivar; therefore, obtaining knowledge of the anthocyanic identity of each variety can provide a tool for applying the most appropriate agronomic and oenological techniques to maximize the varietal expression of the produced wines.

The determination of the anthocyanin content of grapes is a critical topic of the agricultural sector. Their bioactive content could be used as a chemical fingerprint in authenticity studies. Thus, there is a prominent need for the development of efficient, sensitive, and cost-effective analytical methodologies that could be applied in the determination of anthocyanins in grapes. In the literature, liquid chromatographic methods coupled to various detectors such as diode array (DAD) [13], fluorescence (FLD) [14], and mass spectrometric detectors (MS) [15] have been widely used in the analysis of anthocyanins [16]. Sample preparation is the first and most critical step of the analysis process. The applied extraction protocols have already been reviewed [13]. Solid–liquid extraction (SLE) is the most common technique used in the isolation of anthocyanins. According to the literature, extraction is commonly carried out using acetone or acidified methanolic solutions [13,17]. The optimal extraction conditions differ among the various plant materials since the efficiency of the extraction is affected by several parameters such as the type of the matrix, the chemical nature of the sample, the solvent used, the agitation method, the extraction time, the acidifying agent, and the temperature [17,18].

Greece is the cradle of a highly diverse grapevine genetic pool with more than 300 indigenous *V. vinifera* varieties, and most of them are confined in specific geographical areas. To the best of our knowledge, there have been few reports on the individual anthocyanin composition of Greek varieties [19,20]. In this study, we optimized the critical parameters of the extraction protocol for the investigation of the anthocyanin composition of three red-skinned indigenous varieties originating from different areas of Greece, namely ‘Kotsifali’, cultivated in the area of Heraklion in Central Crete island, ‘Vradiano’, cultivated in Evia (the second biggest island of Greece) in Central Greece, and ‘Limnio’, an ancient variety of North Aegean Sea cultivated in the island of Limnos and in the peninsulas of Chalkidiki in North Greece. To achieve this goal, a novel HPLC-DAD method was developed and validated for the determination of anthocyanins.

## 2. Results and Discussion

### 2.1. Optimization of Extraction Parameters

Different extraction systems have been proposed for the isolation of anthocyanins from the matrix [21,22,23]. Ultrasound-assisted extraction (UAE), microwave-assisted extraction (MAE), and supercritical fluid extraction (SFE), among others, have been reported for the extraction of anthocyanins from plant matrices. Among them, UAE has gained popularity owing to its high efficiency, rapidity, and low solvent consumption [24]. Even though UAE has been shown to increase the extraction yield of anthocyanins, it is difficult to establish a consistent extraction protocol for all the plant matrices since these analytes exist in various concentrations depending on the species/cultivar [25] and thus the extraction parameters have to be adjusted to the matrix under study. According to the literature, acetone has resulted in higher recoveries of anthocyanins [26]. The use of aqueous mixtures of acidified solvents has been shown to stabilize the anthocyanins [27]. In order to find the optimum parameters to increase the efficiency of the extraction, the most important factors such as the acidity of the solvent and the temperature during the extraction were optimized [17] using the One Variable at a Time (OVAT) approach [28]. For the optimization experiments, approximately 14 mg of freeze-dried grape skin was weighted in 2-mL dark Eppendorf tubes according to Pinasseau et al. [21]. During the extraction, the efficiency of the following mixtures was evaluated: (a) 0.05% trifluroacetic acid (TFA) in acetone:water (70:30, *v*/*v*); (b) 0.1% TFA in acetone:water (70:30, *v*/*v*); (c) 0.05% HCl in acetone:water (70:30, *v*/*v*); (d) 0.1% HCl in acetone:water (70:30, *v*/*v*). The extraction system with the highest recovery rate was selected as the optimum. The second parameter that was optimized was the extraction temperature after evaluating the efficiency of the extraction at 4 °C, 25 °C, and 60 °C.

#### 2.1.1. Acidifying Solvent

The effects of acidifying the extraction solvent with 0.05% TFA, 0.1% TFA, 0.05% HCl, and 0.1% HCl were evaluated. According to the results presented in Table 1, the calculated recoveries ranged between 89.3 and 116.2%, showing that all the acidifying solvents tested could be successfully used for the extraction of anthocyanins from the grapes. The efficiency of using HCl as an acidifying solvent has already been shown [24,29,30]. The low pH of the extraction favors anthocyanins’ extraction [26], and the findings of this work suggest that a pH over the range 1–2 is satisfactory for the extraction of anthocyanins both for HCl and TFA. To prevent degradation via hydrolysis, 0.05% TFA was chosen as the optimal acid for the extraction.

#### 2.1.2. Extraction Temperature

Another critical parameter that has been demonstrated to affect the anthocyanin extraction yield is the temperature. Three extraction temperatures (4 °C, 30 °C, and 50 °C) were tested, and the extraction recoveries of the anthocyanins are presented in Table 2. According to literature data, the elevated temperatures during the extraction increase the extraction yield, and 50 °C has been selected as the most appropriate temperature in many works; however, this was not the case in our work. Interestingly, the proposed extraction protocol is independent of the extraction temperature, as the extraction recoveries were acceptable, ranging between 88.5% and 104.9% in all cases [17]. The results showed that the temperature does not affect the extraction efficiency. To avoid decomposition [31], the lowest extraction temperature of 4 °C was selected as the optimum to conduct the experiments.

### 2.2. Method Validation Results

The optimized HPLC-DAD methodology was validated to assess the anthocyanin content of the grape skins, and all the analytical parameters, including the calibration curves, linear range, the coefficients of determination (r^2^), accuracy and precision, limits of detection (LODs), and limits of quantification (LOQs), are presented (Table 3, Table 4 and Table 5). The calibration curves were all linear over the range LOQs—20 mg/kg with an r^2^ above 0.99, proving that they can be used for the quantification of the anthocyanins. The LOQs were found to range between 0.20 mg/kg to 0.60 mg/kg, while the LODs were relatively low, with a range from 0.06 mg/kg to 0.20 mg/kg (Table 3). The RSD% of the within-day (*n* = 6) and between-day assays (*n* = 3 × 3) were lower than 6.2% and 8.5%, respectively, showing adequate precision. The accuracy was assessed by means of the relative percentage of recovery (%R) at low, medium, and maximum concentration levels of 0.5, 5, and 20 mg/kg, and the results were acceptable over a range of 91.6–119% for the within-day assay (*n* = 6) (Table 4) and from 89.9 to 123% for the between-day assay (*n* = 3 × 3) (Table 5).

### 2.3. Grape Sample Analysis

#### 2.3.1. Identification of Anthocyanins

Sixteen grape samples belonging to the varieties ‘Kotsifali’, ‘Limnio’, and ‘Vradiano’ were analyzed in triplicate (*n* = 3), and five anthocyanins were determined in all samples. The retention times (RTs) as well as the maximum wavelengths (λmax, nm) for each compound are reported in Table 6. Figure 1 presents a characteristic chromatogram of a real spiked sample at a 1 mg/kg concentration level. Appendix A details the comparison of the different non-spiked chromatograms of the studied grape varieties. All the anthocyanins appeared in all of the studied samples with notable differences in terms of the concentration but also the ratio between them as a characteristic of each variety.

#### 2.3.2. Quantification Results

Significant differences in anthocyanin content were found between varieties, as each one has a specific set of anthocyanins that characterizes it [32]. This heterogeneity is mainly due to the effect of genotype [20]. In total, five anthocyanins were separated and quantified by the HPLC method in all samples, namely the glycosylated derivatives of delphinidin (Dlp), cyanidin (Cyn), petunidin (Pt), peonidin (Pn), and malvidin (Mlv). Apart from the five standard anthocyanins, another two peaks were detected. According to available literature [33,34,35,36], as the corresponding standards were not available, these two compounds should be malvidin-3-O-glucose acetate and p-coumarate, respectively.

Malvidin was by far the predominant anthocyanin in ‘Limnio’ grape samples, which is in agreement with several published studies about indigenous Greek varieties [4,7,37,38]. The quantitative determination of ‘Kotsifali’ and ‘Vradiano’ varieties showed that Mlv was equally important with Pn (Figure 2). Moreover, in the study of the Portuguese variety Alvarilhão [34], Mlv had a similar content to Pn. Two of the three varieties had comparable levels of Mlv (‘Kotsifali’ and ‘Vradiano’), while the ‘Limnio’ variety appeared with up to two-times higher concentration (8.25–12.7 mg/100 g fresh weight (f.w.) in ‘Kotsifali’, 6.94–23.2 mg/100 g f.w. in ‘Limnio’, and 6.15–10.1 mg/100 g f.w. in ‘Vradiano’ grapes). According to the total anthocyanin quantification results that are graphically illustrated in Figure 3, the highest total anthocyanin concentration was observed in grape samples belonging to the ‘Kotsifali’ variety (49.2 mg/100 g f.w.), while the lowest was observed in samples belonging to the ‘Limnio’ variety (8.92 mg/100 g f.w.).

Upon analyzing each monoglucoside separately, Cyn exhibited the lowest mean content (0.33 mg/100 g f.w. for ‘Limnio’ and 0.81 mg/100 g f.w. for ‘Vradiano’) followed by Dlp (0.60 mg/100 g f.w. in ‘Limnio’ and 0.87 mg/100 g f.w. in ‘Vradiano’). The exception was the variety of ‘Kotsifali’, which presented a remarkably high concentration of Cyn when compared to the other varieties (4.93 mg/100 g f.w.). On the contrary, in ‘Limnio’ grapes, all anthocyanins except of Mlv had a small contribution to the total pool of anthocyanic content, while ‘Vradiano’ displayed an intermediate profile. All of the above are presented in detail in Table 7. These results are in agreement with previous findings for some of these varieties [36,39], except for ‘Vradiano’, a rare red grape, which has never been analyzed previously.

Relating the above results to percentages of each anthocyanin sum per variety (Figure 4), many notable differences emerged between the three varieties. Specifically, ‘Limnio’ skins’ Mlv, which is the most stable of the five anthocyanins [40], was present at a range of 58–78.1% of the total. The respective percentage for ‘Kotsifali’ was 25.8–43%, while, for ‘Vradiano’, it was 38.1–49.9%. In both ‘Kotsifali’ and ‘Vradiano’, the contribution of Mlv was in the same level with Pn (32.4–36.5% in ‘Kotsifali’ and 37.3–48.8% in ‘Vradiano’). The ‘Kotsifali’ variety also stood out due to its high percentage of Cyn (7.73–20.4%) when compared with the other studied varieties, wherein the latter anthocyanin was detected in traces. These results are in agreement with previous studies for this variety [4,20]. As mentioned previously, methoxylated anthocyanins, such as peonidin and malvidin, are more stable [20]. Consequently, the variety of ‘Kotsifali’, although rich in anthocyanins, is expected to have unstable color in the wines due to a high concentration of Cyn. The opposite is true for ‘Limnio’, with less but more stable color.

These differences in both concentration and types of anthocyanin can lead to differences in intensity, hue, and color stability overall. Therefore, the knowledge of the anthocyanin identity of each variety can be a tool for the application of the most appropriate agronomic and oenological techniques to maximize the varietal expression of the produced wines.

### 2.4. Hierarchical Cluster Analysis

Hierarchical cluster analysis (HCA) was performed on the data matrix of 16 samples × 5 anthocyanins without a priori knowledge about the group structure of the dataset, measuring the distance between each pair of objects in terms of variables and grouping the objects that are close. HCA was applied to produce a tree diagram and identify the groups with objects of a high degree of similarity. The algorithm starts by treating each object as a singleton cluster (leaf); then, pairs of clusters are merged until all clusters have been successively merged into one large cluster that contains all the objects, resulting in a dendrogram.

The heatmap and the developed dendrogram of the HCA are presented in Figure 5, showing the clustering of three major groups, one for each variety.

Following the importance score, the A3 sample stands out due to its increased relativity to all of the monoglycoside anthocyanins. Similar behavior was displayed in all of the ‘Kotsifali’ samples except for the A5 sample, which was distinguished from the others by its high concentration at Cyn and low correlation with the others anthocyanins. The above results for the ‘Kotsifali’ variety, with the anthocyanins of Pn and Cyn standing out overall, are also confirmed in the heat map and the clustering of this variety. The anthocyanin composition of the variety ‘Limnio’ consists almost exclusively of Mlv, a fact confirmed by the dendrogram (Figure 5), with samples from the main cultivation areas of the variety (B6, B3, and B4) classified in the same sub-cluster. Lastly, Pn had an important percent in ‘Vradiano’ cultivar (37.3–48.8%) (Appendix A), which is characteristic of the variety and also becomes distinct in the heatmap (Figure 5). HCA analysis of the results showed (Figure 5) separation of the three varieties, particularly ‘Kotsifali’, from ‘Limnio’. The ‘Vradiano’ variety differentiates, however, in the presence of the A5 sample because of Cyn concentration.

With the exception of A5, our results indicate that grape skin anthocyanin concentration ranges are characteristic of each variety and can therefore be used as a chemical indicator to distinguish Vitis vinifera varieties. Many studies using chemometric methods and especially HCA are able to classify wine and grapes according to grape variety, the phenolic content, or another variable. HCA analysis has been used to separate the cultivars and sort them based on skin color [41]. Clustering analysis has been employed for the discrimination of the Greek red wines belonging to the varieties ‘Kotsifali’ and ‘Mandilaria’ [42]. In another work, HCA was used for the clustering of red wines from China using the concentration levels of individual phenolic compounds [43].

## 3. Materials and Methods

### 3.1. Samples Collection

Grape samples belonging to the indigenous red grape of ‘Kotsifali’, ‘Limnio’, and ‘Vradiano’, originating from Greece, were collected at the stage of optimum maturity during the harvesting period of 2020. A representative sample was taken at the same time on the day for each cultivar from different vineyards throughout Greece depending on availability. Details about the variety and the geographical origin of the samples are presented in Table 8. Samples of 50 berries were collected for the grape maturity analysis. For the analysis of anthocyanins, another 50 berries were collected from each vineyard to form a bulk sample. Grape samples for anthocyanin analysis were put in boxes with dry ice and were brought to the laboratory and stored in a deep freezer (−80 °C) until further treatment.

### 3.2. Sample Preparation

Most of 50 grapes were used for the calculation of total soluble solids (Brix), total titratable acidity (TA), and pH. These measurements are a part of grape maturity analysis (Appendix A). The samples for the anthocyanin analysis were weighted, and the skins were manually isolated. The skins were freeze-dried for two days and then were ground to obtain powder. The pulverized freeze-dried skins were stored in a deep freezer (−25 °C) until the analysis.

### 3.3. Reagents and Standards

Methanol and acetone (HPLC grade) were purchased from Merck (Zedelgem, Belgium). Methanol (HPLC-Ultra LC-MS) from HiPerSolv CHROMANORM, VWR Chemicals BDH (The Netherlands), was also purchased for the production of standard solutions. Formic acid (99%) and hydrochloric acid (37%) for analysis were purchased from Carlo Erba (Chaussée du Vexin, France), and trifluoroacetic acid for LC-MS was obtained from Fluka (Buchs, Switzerland). Standards of delphinidin-3-O-glucoside chloride, cyanidin-3-O-glucosidechloride, petunidin-3-O-glucoside chloride, peonidin-3-O-glucoside chloride, and malvidin-3-O-glucoside chloride were obtained from Extrasynthese (Genay Cedex, France). The standards of delphinidin-3-O-glucoside (Dlp), cyanidin-3-O-glucoside (Cyn), petunidin-3-O-glucoside (Pt), peonidin-3-O-glucoside (Pn), and malvidin-3-O-glucoside (Mlv) were diluted separately in methanol LC-MS with 0.1% HCl (concentration 1000 mg/L).

### 3.4. Instrumentation

Chromatographic analysis was carried out in a SpectraSYSTEM (Thermo Separation Products, Austin, TX, USA) HPLC system consisting of a P2000 secondary solvent pump, an AS3000 autosampler equipped with a 100-μL injection loop and a UV6000LP diode array detector. Chromatographic data were monitored and processed by ChromQuest 5.0 (Thermo Fisher Scientific Inc., Waltham, MA, USA). The samples were freeze dried in an Alpha 2–4 LD freeze dryer acquired from Martin Christ Gefriertrocknungsanlagen GmbH (Osterode am Harz, Germany) which was equipped with a two-stage vacuum rotary pump RZ 2.5 Vacuubrand (condenser temperature: −80 °C, max flow 2.3/2.8 m^3^ h^−1^, ultimate vacuum 4 × 10^−4^ mbar). A 5804 R centrifuge system with rotor F-45-30-11 was acquired from Eppendorf AG (Germany). Water was purified in a Direct-Q^®^ 3 UV Water Purification System acquired from Merck KGaA, Darmstadt, Germany. For filtering the aqueous mobile phase, ME 25 ST 0.45-µm membrane filters (Schleicher and Schuell, W. Germany) were used. For solvent evaporation under nitrogen gas, a TurboVap LV workstation was used by Caliper Life Sciences (Hopkinton, MA, USA). An ultrasonic bath RK 100H (Bandelin Sonorex, Berlin, Germany) and a Stuard-SB3 stirrer were used for the extraction. Regenerated Cellulose 0.22 μm (RC) syringe filters (Captiva, Agilent Technologies, Santa Clara, CA, USA) were used for filtering the samples.

### 3.5. Extraction of Anthocyanins and Anthocyanidins

Freeze-dried skin was weighted (0.0140 g) and extracted with 200 μL of methanol and 1.4 mL of acetone/water/TFA (70:29.95:0.05). The solution was extracted under sonication for 10 min and stirred at 40 rpm for 20 min, repeated twice. The extraction was carried out at 4 °C in the absence of light and the extract was centrifuged and thermostated at 4 °C, 10,000 rpm, for 10 min. Then, a fraction of 0.5 mL of the supernatant was dried with nitrogen under pressure. At this stage, and unless chromatographic analysis was followed immediately, the samples were stored as concentrates in the deep freezer (−80 °C). The concentrate was redissolved with a mixture of 250 μL methanol and 750 μL water 0.134% formic acid. Subsequently, the solution was led to an ultrasonic device again at 4 °C in the dark for 30 min. The extract was centrifuged and thermostated at 4 °C, for 14,000 rpm, for 15 min, and the supernatant was filtered through 0.22-μm RC syringe filters (RC) (Captiva, Agilent Technologies) prior to chromatography.

### 3.6. HPLC-DAD Analysis

Τhe analysis of anthocyanins was performed using a modified chromatographic method according to Kyraleou et al. [22]. Chromatographic separation was performed on a Nucleosil 100-5 C_18_, 250 × 4.6 mm, 5-μm, reversed-phase (RP) column (Macherey–Nagel, Düren, Germany). The DAD detector was set over the range 500–550 nm. The column oven temperature was 40 °C, the injection volume was 5 μL, and the total runtime was 40 min. The mobile phases were aqueous formic acid 5% (Solvent A) and methanol (Solvent B) at a flow rate of 1 mL/min. The gradient composition is presented in Table 9.

### 3.7. Method Validation

The method validation was performed to estimate linearity, selectivity, LODs and LOQs, trueness, and precision. Linearity studies were performed in triplicate and covered the entire working range. The calibration curves of anthocyanins were constructed by plotting the peak area versus concentration. LODs were calculated as three signal to noise ratios (3 S/N), and the formula LOQ = 10 S/N was employed for the calculation of the LOQ [44]. Trueness and precision were studied using real grape skin samples spiked at three different concentrations (0.5, 5, and 20 mg/kg) and were analyzed in triplicate. To evaluate trueness, relative recoveries (%R) were calculated by means of recovery percentage by comparing the found and added concentrations of the examined analytes (mean concentration found/added concentration × 100). The precision of the method was expressed in terms of relative standard deviation (RSD%) and was calculated for repeated measurements of spiked samples. Following this approach, within-day precision (repeatability) was assessed in five replicates, while between-days precision (reproducibility) was assessed by performing triplicate analysis for spiked samples within four consecutive days [44]. In order to assess selectivity, five blank matrices were used, and no interferences were observed in the same chromatographic window as the anthocyanins examined.

### 3.8. Chemometric Analysis

The statistical differences between the species on the basis of their elemental concentration were estimated with ANOVA at a 95% confidence level (*p* < 0.05) in Microsoft Excel ((Microsoft, WA, USA) using the Data Analysis tool. The anthocyanin concentrations in grape samples were analyzed by hierarchical cluster analysis (HCA), which is a suitable method for small quantities of data, using the average between-groups linkage method and squared Euclidean distance interval measurement [45].

## 4. Conclusions

The purpose of this study was two-fold: first, to develop a simple and efficient methodology for the identification and quantification of grape skin anthocyanin in red cultivars, and, second, to investigate the anthocyanin profile of three Greek indigenous winegrape varieties as a means to distinguish cultivars based on chemometric analysis. For the preparation of the red grape skins to be analyzed, a rapid pretreatment protocol of the grape extracts was chosen, which combines simple and short purification techniques with a positive environmental footprint, due to the use of minimal amounts of solvents. The extraction protocol developed offers the possibility of the rapid identification and quantification of anthocyanins in order to characterize red varieties according to their anthocyanin profile. The grapevine variety ‘Kotsifali’ appeared richer in anthocyanins, albeit with a greater participation in the anthocyanin profile of the di-hydroxylated anthocyanins Pn and Cyn, while the opposite was observed in the ‘Limnio’ variety. This knowledge can improve the viticultural and oenological management of these varieties. Furthermore, the data revealed valuable information regarding the chemical separation of the wines of the three varieties based on chemometric analysis. These results create the conditions for further investigation in a large number of Greek grape varieties with the objective of establishing reference anthocyanin profiles to distinguish the varieties alone or in cooperation with molecular analyses.

## Figures and Tables

**Figure 1 molecules-27-07107-f001:**
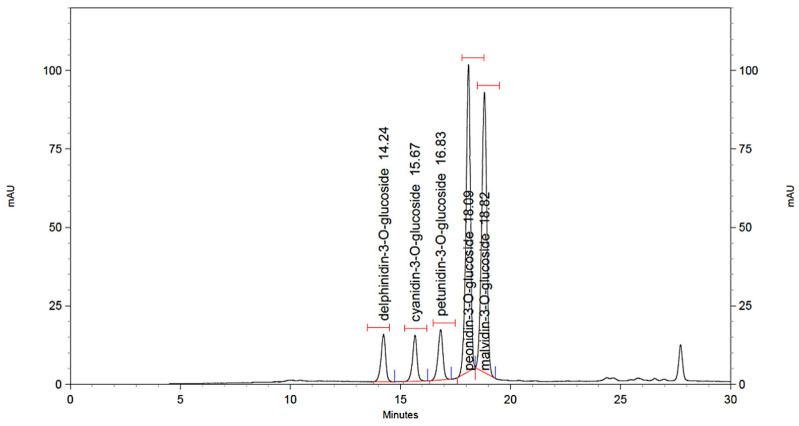
Characteristic chromatogram of a real sample spiked at 1 mg/kg concentration level, monitored at 520 nm.

**Figure 2 molecules-27-07107-f002:**
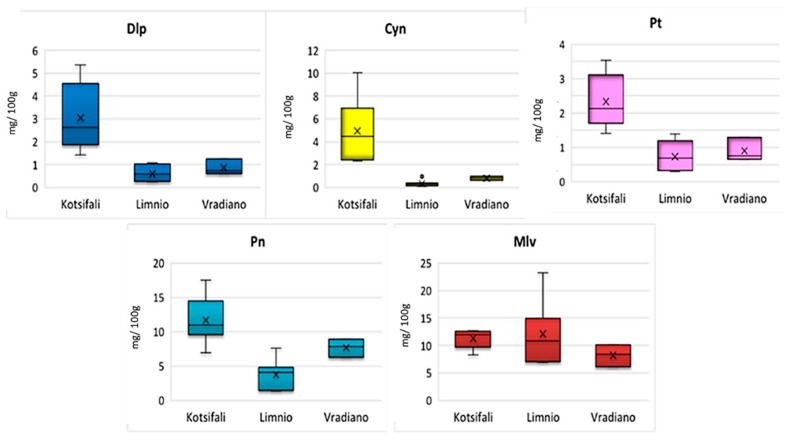
Box and whisker plots for the concentrations of Dlp; Cyn; Pt; Pn; Mlvin red grapes belonging to ‘Kotsifali’, ‘Limnio’, and ‘Vradiano’.

**Figure 3 molecules-27-07107-f003:**
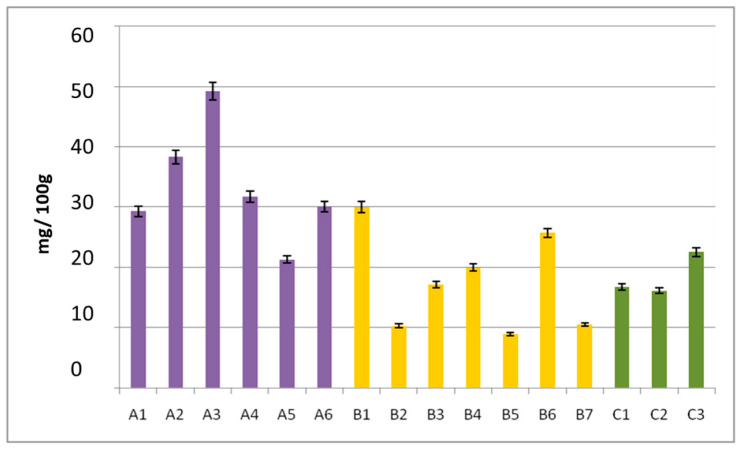
Total anthocyanin content (sum of individual anthocyanins ± SD) in red grapes of ‘Kotsifali’ (marked in purple), ‘Limnio’ (marked in yellow), and ‘Vradiano’ (marked in green).

**Figure 4 molecules-27-07107-f004:**
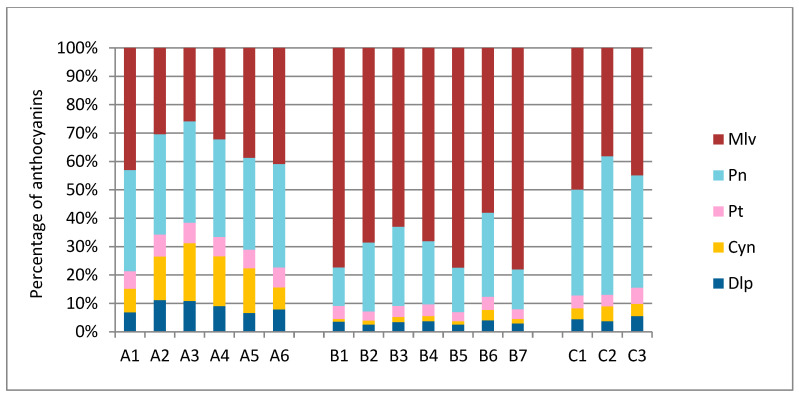
Percentage of individual anthocyanin content of Mlv; Pn; Cyn; Dlp; in the skins of the red grapes belonging to ‘Kotsifali’, ‘Limnio’, and ‘Vradiano’.

**Figure 5 molecules-27-07107-f005:**
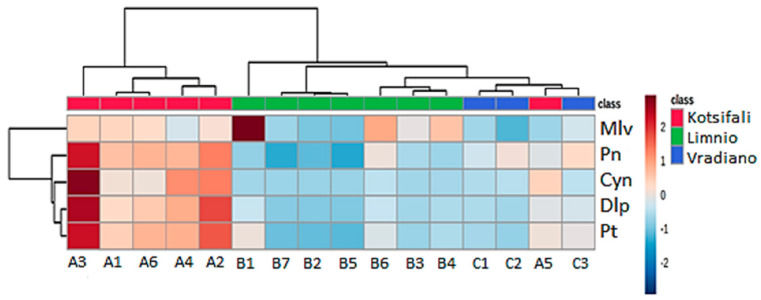
Hierarchical cluster analysis (HCA) of 16 samples of red grapes based on anthocyanins (Mlv; Pn; Cyn; Dlp; Pt).

**Table 1 molecules-27-07107-t001:** Recovery (%) of anthocyanins extracted with different acidified solvents.

Acidifier	%R Dlp	%R Cyn	%R Pt	%R Pn	%R Mlv
**0.05%TFA**	91.0 ± 1.4	91.5 ± 1.0	101.5 ± 14.6	97.7 ± 4.4	99.0 ± 0.2
**0.1%TFA**	91.9 ± 2.0	91.4 ± 1.4	93.1 ± 1.2	99.5 ± 2.1	101.1 ± 3.3
**0.05%HCl**	110.8 ± 14.8	104.9 ± 12.2	111.3 ± 11.7	116.2 ± 16.9	114.8 ± 13.6
**0.1%HCl**	89.3 ± 5.8	87.5 ± 2.7	95.4 ± 3.5	96.3 ± 0.2	98.5 ± 2.8

Dlp: delphinidin-3-O-glucoside; Cyn: cyanidin-3-O-glucoside; Pt: petunidin-3-O-glucoside; Pn: peonidin-3-O-glucoside; Mlv: malvidin-3-O-glucoside

**Table 2 molecules-27-07107-t002:** Recovery (%) of anthocyanins extracted at different temperatures.

Temperature	Dlp	Cyn	Pt	Pn	Mlv
**4 °C**	97.9 ± 1.3	101.6 ± 1.0	102.6 ± 4.7	93.8 ± 4.4	99.8 ± 0.2
**30 °C**	90.0 ± 6.6	91.6 ± 5.2	88.5 ± 7.6	96.4 ± 2.2	91.7 ± 5.4
**50 °C**	98.6 ± 0.9	104.9 ± 3.0	99.2 ± 0.6	104.0 ± 2.2	99.1 ± 0.6

Dlp: delphinidin-3-O-glucoside; Cyn: cyanidin-3-O-glucoside; Pt: petunidin-3-O-glucoside; Pn: peonidin-3-O-glucoside; Mlv: malvidin-3-O-glucoside.

**Table 3 molecules-27-07107-t003:** HPLC-DAD validation parameters.

Compound	Calibration Equation	Linear range (mg/kg)	r^2^	LOD (mg/kg)	LOQ (mg/kg)
**Dlp**	y = 72,031.3x − 8267	LOQ-20	0.996	0.12	0.40
**Cyn**	y = 77,893.6 x − 6868	LOQ-20	0.999	0.10	0.30
**Pt**	y = 73,634.8x + 2305	LOQ-20	0.999	0.18	0.60
**Pn**	y = 68,782.6x + 1053	LOQ-20	0.999	0.06	0.20
**Mlv**	y = 55,442x + 1895	LOQ-20	0.999	0.10	0.30

Dlp: delphinidin-3-O-glucoside; Cyn: cyanidin-3-O-glucoside; Pt: petunidin-3-O-glucoside; Pn: peonidin-3-O-glucoside; Mlv: malvidin-3-O-glucoside.

**Table 4 molecules-27-07107-t004:** Repeatability results of the method estimated as recoveries (%R, *n* = 6) for the studied anthocyanins at three fortification levels.

Compound	Low Concentration (%R, *n* = 3 × 3)	%RSD	Medium Concentration (%R, *n* = 3 × 3)	%RSD	High Concentration (%R, *n* = 3 × 3)	%RSD
**Dlp**	114	3.9	112	7.4	106	4.8
**Cyn**	105	7.5	96.7	7.9	91.6	7.3
**Pt**	113	6.5	111	6.5	106	6.2
**Pn**	119	6.2	118	5.8	107	7.1
**Mlv**	118	8.4	116	7.4	108	5.4

Dlp: delphinidin-3-O-glucoside; Cyn: cyanidin-3-O-glucoside; Pt: petunidin-3-O-glucoside; Pn: peonidin-3-O-glucoside; Mlv: malvidin-3-O-glucoside.

**Table 5 molecules-27-07107-t005:** Intermediate precision results of the method estimated as recoveries (%R, *n* = 3 × 3) for the studied anthocyanins at three fortification levels.

Compound	Low Concentration (%R, *n* = 3 × 3)	%RSD	Medium Concentration (%R, *n* = 3 × 3)	%RSD	High Concentration (%R, *n* = 3 × 3)	%RSD
**Dlp**	120	7.3	114	2.4	111	4.4
**Cyn**	114	5.0	94.7	2.3	89.9	3.8
**Pt**	118	4.7	107	3.1	110	6.1
**Pn**	103	9.6	119	4.1	123	1.4
**Mlv**	115	4.1	120	6.8	117	4.4

Dlp: delphinidin-3-O-glucoside; Cyn: cyanidin-3-O-glucoside; Pt: petunidin-3-O-glucoside; Pn: peonidin-3-O-glucoside; Mlv: malvidin-3-O-glucoside.

**Table 6 molecules-27-07107-t006:** Chromatographic retention times and maximum wavelengths of anthocyanins.

Compound	RT	λ max (nm)
**Dlp**	14.3	516
**Cyn**	15.7	510
**Pt**	16.8	543
**Pn**	18.1	512
**Mlv**	18.8	520

Dlp: delphinidin-3-O-glucoside; Cyn: cyanidin-3-O-glucoside; Pt: petunidin-3-O-glucoside; Pn: peonidin-3-O-glucoside; Mlv: malvidin-3-O-glucoside

**Table 7 molecules-27-07107-t007:** Anthocyanin concentration levels in red grapes of ‘Kotsifali’, ‘Limnio’, and ‘Vradiano’ (samples analyzed in triplicate, *n* =3 ± SD).

Variety	Sample Name	Dlp (mg/100 g)	Cyn (mg/100 g)	Pt (mg/100 g)	Pn (mg/100 g)	Mlv (mg/100 g)	Total (mg/100 g)
**Kotsifali**	A1	2.02 ± 0.04	2.43 ± 0.07	1.80 ± 0.04	10.4 ± 0.21	12.6 ± 0.23	29.3 ± 0.58
	A2	4.27 ± 0.16	5.9 ±0.07	2.96 ± 0.11	13.5 ± 0.32	11.6 ± 0.51	38.3 ± 1.17
	A3	5.35 ± 0.15	10.0 ± 0.11	3.54 ± 0.17	17.6 ± 0.50	12.7 ± 0.66	49.2 ± 1.59
	A4	2.88 ± 0.14	5.55 ± 0.33	2.16 ± 0.04	10.9 ± 0.13	10.2 ± 0.29	31.7 ± 0.92
	A5	1.42 ± 0.08	3.35 ± 0.13	1.41 ± 0.07	6.91 ± 0.26	8.25 ± 0.20	21.3 ± 0.74
	A6	2.38 ± 0.06	2.32 ± 0.11	2.12 ± 0.03	11.0 ± 0.17	12.3 ± 0.26	30.1 ± 0.72
**Limnio**	B1	1.07 ± 0.06	0.29 ± 0.01	1.39 ± 0.07	4.07 ± 0.20	23.2 ± 0.89	30.0 ± 1.23
	B2	0.26 ± 0.02	0.15 ± 0.01	0.33 ± 0.03	2.51 ± 0.17	7.08 ± 0.40	10.3 ± 0.64
	B3	0.59 ± 0.03	0.30 ± 0.01	0.68 ± 0.03	4.78 ± 0.11	10.8 ± 0.55	17.2 ± 0.71
	B4	0.74 ± 0.14	0.37 ± 0.05	0.84 ± 0.14	4.44 ± 0.57	13.6 ± 1.83	20.0 ± 2.70
	B5	0.23 ± 0.01	0.10 ± 0.01	0.29 ± 0.01	1.40 ± 0.12	6.94 ± 0.18	8.97 ± 0.32
	B6	1.03 ± 0.04	0.96 ± 0.16	1.19 ± 0.04	7.62 ± 0.62	14.9 ± 0.89	25.7 ± 1.75
	B7	0.31 ± 0.01	0.17 ± 0.01	0.37 ± 0.01	1.47 ± 0.06	8.22 ± 0.28	10.5 ± 0.36
**Vradiano**	C1	0.75 ± 0.02	0.64 ± 0.02	0.76 ± 0.02	6.25 ± 0.09	8.35 ± 0.17	16.7 ± 0.31
	C2	0.60 ± 0.03	0.84 ± 0.02	0.66 ± 0.03	7.88 ± 0.36	6.15 ± 0.25	16.1 ± 0.69
	C3	1.25 ± 0.16	0.96 ± 0.05	1.30 ± 0.16	8.90 ± 0.21	10.1 ± 0.64	22.5 ± 1.21

Dlp: delphinidin-3-O-glucoside; Cyn: cyanidin-3-O-glucoside; Pt: petunidin-3-O-glucoside; Pn: peonidin-3-O-glucoside; Mlv: malvidin-3-O-glucoside.

**Table 8 molecules-27-07107-t008:** Variety and geographical origin of the samples.

Grape Variety	Sample No.	Geographical Origin	Location
**Kotsifali**	A1	Crete	Katw Asites
	A2	Crete	Dafnes
	A3	Crete	Alagni-Peza
	A4	Crete	Arxanes
	A5	Attica	Wine Institute
	A6	Macedonia	Greek Genebank
**Limnio**	B1	Attica	Wine Institute
	B2	Macedonia	Epanomi
	B3	Macedonia	Mount Athos
	B4	Macedonia	Sithonia
	B5	Thrace	Xanthi
	B6	Aegean Sea	Limnos
	B7	Macedonia	Serres
**Vradiano**	C1	Attica	Wine Institute
	C2	Evia	Istiaia
	C3	Evia	Gialtra

**Table 9 molecules-27-07107-t009:** HPLC Gradient for the separation of anthocyanins in grape skin extracts.

Time (Min)	Solvent A (%)	Solvent B (%)
0	90	10
22	50	50
32	5	95
34	5	95
35	90	10
40	90	10

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
