# Peer review of "An Optimized HPLC-DAD Methodology for the Determination of Anthocyanins in Grape Skins of Red Greek Winegrape Cultivars (*Vitis vinifera* L.)"

_molecules, 2022, doi:10.3390/molecules27207107_

Round 1

Reviewer 1 Report

The Authors of the work submitted for review described a simple HPLC-DAD method for determining five anthocyanins in grape skins. They presented the optimization of the extraction process and discussed the obtained results. Hierarchical cluster analysis was used to visualize the differences between the 16 samples from the three cultivars from 14 locations. The work is interesting; all stages of the procedure are clearly presented, and the optimization of the sample preparation method and the validation of the chromatographic method are appropriate. The discussion of the results with the graphs is clear, and the conclusions are correct. I think the work should be published in Molecules; I believe that the article may interest readers not only dealing with wine production.

I noticed some editorial errors:

Related to the lack of space before the units: in the abstract before mg/kg; in section 2.1.2. before °C; in section 2.3.2 before g,

-  In line 131 at the% sign. It is 116.2% should be 116.2%

-  The numbering in the list of references has been duplicated.

Author Response

Response: We thank the reviewer for his/her nice words and the helpful comments.

Comment 1: Related to the lack of space before the units: in the abstract before mg/kg; in section 2.1.2. before °C; in section 2.3.2 before g,

Response: It was corrected in the revised manuscript.

Comment 2: In line 131 at the% sign. It is 116.2% should be 116.2%

Response: It was corrected in the revised manuscript.

Comment 3: The numbering in the list of references has been duplicated.

Response: It was corrected in the revised manuscript.

Reviewer 2 Report

In this work, a simple and classical liquid-solid extraction method to determine 5 anthocyanins in grape skin was developed. The manuscript is well organized, the experiments were properly done and the results well interpreted also. The English grammar is very good. All Tables and Graphics are clear. The analytical procedure was internally validated obtaining very good figures of merit, however, the authors should compared those figures (mainly the LODs) with other procedures from the literature to determine the studied anthocyanins. Some minor and other important points should be addressed before publication:

Abstract

Line 21: changed by 6.2 %, and 8.5%. The same in line 169.

Section 2.1: define the acronym OVAT, since it was not done before. The kinf of extraction procedure should be mentioned in the Abstract. Why the ratio acetone/water was not varied and optimized?

Line 145: has been shown…

Line 163: replace by “coefficient of determination”

 The LOQs and LODs should be compared with other similar procedures in the literature.

A chromatogram for a non-spiked sample should also be shown. It would be very interesting a comparison of the different chromatograms for the native samples of the different studied grape varieties.  

In Section 3.6 the authors mention that the detector was set at 520 nm but in Table 6 the maximum wavelengths are shown for each compound, which was the purpose of this?

All the 5 compounds elute in a time range of about 7 min, when the gradient is in about 40 and 50% methanol. This is not surprising since all compounds have similar structures. Why the gradient was necessary? This is a typical chromatographic situation in which a few compounds eluting together in a gradient can be resolved in an isocratic mode (e.g. 40 % methanol) in a shorter time, because the first part of the gradient has a high content of water, delaying the elution (40 min total analysis time to elute 5 compounds is a quite long time analysis). Time analysis is very important in this work since the amount of sample to be analyzed is very high. Also, to shorten the analysis time, a shorter column should be used, such as 15 cm. All of this, make the method greener, since less mobile phase will be used.

Line 405: performing

Conclusions section: In order to evaluate the greenness of the method, some procedures such as GAPI or AGREE should be used. The sample preparation requires minimum organic solvents in this work but other factors such as analytical technique, amount of mobile phase and general instrumentation should be evaluated. Actually, it is a global tendency of all analytical community to add the word “green” to the manuscript of a new developed analytical method but the evaluation is subjective. This reviewer considers that, if no metrics about the greenness of the method was done, is not proper to qualify it as “green” just because a few amounts of organic solvents were used in the sample preparation step.

Line 421: Why the developed method was qualified as “modern”? How were the old procedures?

Ckeck reference numbers, they are repeated.

Author Response

Response: We are grateful to the reviewer for his/her very kind comments.

Please find all the responses in the attached file
